# Probiotics and the Potential of Genetic Modification as a Possible Treatment for Food Allergy

**DOI:** 10.3390/nu15194159

**Published:** 2023-09-27

**Authors:** Yuqiu Wei, Jing Peng, Siyu Wang, Zheng Ding, Guixi Chen, Jiazeng Sun

**Affiliations:** 1Key Laboratory of Precision Nutrition and Food Quality, Department of Nutrition and Health, China Agricultural University, Beijing 100190, China; 2College of Biological Sciences, China Agricultural University, Beijing 100193, China

**Keywords:** probiotics, food allergy, genetic modification, precision therapy

## Abstract

Food allergy is a common condition that affects millions of people worldwide. It is caused by an abnormal immune response to harmless food antigens, which is influenced by genetics and environmental factors. Modulating the gut microbiota and immune system with probiotics or genetically modified probiotics confers health benefits to the host and offers a novel strategy for preventing and treating food allergy. This systematic review aims to summarize the current proof of the role of probiotics in food allergy and propose a promising future research direction of using probiotics as a possible strategy of treatment for food allergy.

## 1. Introduction

Food allergy (FA) has gradually become one of the most concerning public health problems around the globe over the past decades. FA is a hypersensitivity reaction caused by the immunological response to the exposure of a specific food allergen. This can lead to symptoms such as urticaria, gastroenteritis, angioedema, and anaphylaxis, which can potentially be fatal. It is worth mentioning that an FA is different from a food intolerance, which does not involve the immune system and is usually less severe. The prevalence of FA varies by culture and region due to several factors, including genetic, environmental, and dietary differences. Reported statistics show that approximately 4.3% of the global population is affected by FA. In China, the prevalence of self-reported and hospital-diagnosed FA from 2009 to 2018 was 8%, presenting a rising trend compared to the data from 2001 to 2009, which was 5%. According to reported data, the most common food allergens globally were cow’s milk, egg, peanuts, tree nuts, shellfish, fish, wheat, and soybeans. However, specific FAs vary from culture to region as dietary customs change. For instance, rice allergy is more common in Asia in comparison to North America [1,2].

Depending on the mechanism, food allergies can be IgE-mediated (immediate response), non-IgE-mediated (delayed response), or a combination of both [3]. The current option for the management of FA is based on the avoidance of the consumption of food allergens and treatment of accidental reactions with antihistamines, corticosteroids, or epinephrine [4]. However, many patients are not satisfied with the outcomes, as they compromise their quality of life and expose them to a high risk of anaphylaxis. Therefore, there is a need for alternative or complementary approaches to prevent and treat FA. Probiotics, which are live microorganisms that provide a health benefit to the host when given in adequate amounts, are one of the possible candidates. Probiotics may modulate the gut microbiota, which is essential for the development and regulation of immune tolerance. Probiotics may also affect the systematic immune response by altering the production of cytokines, antibodies, and regulatory T cells [5]. The effects of probiotics on FA have been explored in several studies, either as a prevention strategy in high-risk infants or as a treatment strategy in patients with established FA [4,5]. However, the findings are inconclusive and contradictory, possibly due to the variability of probiotic strains, doses, durations, formulations, and outcomes. Furthermore, the safety and quality of probiotic products are not well controlled, and some adverse effects have been reported.

The terms “synthetic biology” and “precision therapy” have become popular over the past decades, with their dedications as treatments for various diseases including cancer. Synthetic biology works with the smallest possible components of living systems, such as DNA, proteins, and other organic molecules, and tries to build fully functional biological systems [6]. Precision therapy aims to deliver the right treatments to the right patients at the right time, and to enhance the efficacy and safety of the therapies. Precision therapy is also sometimes referred to as personalized medicine or personalized care [7]. These might be two of the potential fields of research scientists can explore and apply to the treatment and prevention of FA using probiotics.

Therefore, this review summarizes the current research on probiotics intervention and treatment of FA and aims to provide inspiration on the possible future research direction of using probiotics as a strategy treatment for FA.

## 2. The Mechanism of IgE-Mediated Food Allergy

IgE-mediated FA is the most fully characterized type due to its association with more severe and possibly fatal reactions, and thus is usually considered to be the most serious type of FA. IgE-mediated FA is a type I hypersensitivity reaction driven by the shift of balance towards T helper type 2 (T_H_2) immunity versus T_H_1 immunity. The adaptive immune system’s T_H_2 branch promotes CD4+ T_H_2 cells, eosinophils, basophils, mast cells, type 2 innate lymphoid cells, and inflammatory cytokines such as IL-4, IL-5, IL-9, and IL-13, as well as IgE-mediated humoral antibody responses [8].

The mechanism involves three stages: sensitization, elicitation, and inflammation [9]. Sensitization refers to the stimulation of IgE antibodies by B cells following the initial exposure to a food antigen. B cells perform one of the critical roles in the mechanism of sensitization—the release of immunoglobin through class switch recombination (CSR) [10]. CSR is a sophisticated and strictly regulated process; when a naïve B cell is stimulated by different cytokines and co-stimulating factors, the variable region on the immunoglobin heavy chain will undergo genetic recombination and therefore express different types of immunoglobin in specific situations. In the case of FA, IgE is produced as a result.

After the release, IgE antibodies will bind to high-affinity IgE receptors (FcεRI) expressed on immune effector cells, including mast cells and basophils, which contain granules with inflammatory mediators like histamine [8]. Sensitization may be asymptomatic and may occur early or late in life. Elicitation is the re-exposure to the same food antigen that triggers the IgE antibodies bound to mast cells and basophils to cross-link, leading to the degranulation and release of other substances that initiate an allergic reaction (Figure 1). The elicitation phase results in symptoms that vary from mild to severe and typically occur within minutes to hours of ingesting the food. Inflammation is the consequence of the allergic reaction that engages various immune cells, cytokines, chemokines, and other molecules that enhance and maintain the inflammatory response. The inflammation phase can induce tissue damage and chronic complications such as eosinophilic esophagitis or oral allergy syndrome [3,9].

## 3. The Relationship of the Gut Microbiota and Food Allergy

The gut microbiota is the collection of microorganisms that live in the human digestive tract and influence various aspects of health and disease [11]. The gut microbiota consists of trillions of microbes, including a diverse variety of bacteria, fungi, viruses, and archaea that inhibit the human intestines and form a complex ecosystem [12]. This complex ecosystem can be influenced by a range of factors, such as genetics, diet, environment, and medications (Figure 2). The microbial composition can affect the risk and progression of numerous diseases, including FA. Several studies have shown the composition of the gut microbiota is intensely correlated with the development of FA [11,12,13,14,15,16,17].

A review by M. Tanaka and J. Nakayama [14] gathered the current knowledge on the relationship between the development of the gut microbiota and its impact on one’s health later in life, and discussed how the gut microbiota develops in the first year of life and how it influences the maturation of the digestive tract and the immune system, as well as the risk of allergic and autoimmune diseases. The study suggested that the gut microbiota is tightly connected to the healthy living of an individual later in life and has suggested that early intervention with probiotics or prebiotics modulates the gut microbiota and improves infant health. The type and duration of complementary feeding and breastfeeding are the main factors influencing gut microbiota development [14]. The establishment and maturation of the gut microbiota in infancy are intimately linked to immune system development, and deviations from the normal gut microbiota trajectory may predispose to autoimmune or allergic diseases. In addition, multiple studies have shown breastfeeding and vaginal delivery have beneficial effects on the infant’s intestinal microbiota composition and confer protection against allergy development [14,16,18].

Vaginally delivered newborns can gain their first inoculum through the maternal vaginal tract, which contains an abundance of microorganisms. Exposure to such diversity can benefit the infants’ development of gut microbiota and have countless advantages compared to cesarian section (CS) birth [13,18,19]. CS delivery has been proven to be linked to dysbiosis in the gut microbiota and the increased incidence of FA. In murine models, mice delivered by CS were more susceptible to FA to ovalbumin (OVA) than mice delivered vaginally, as demonstrated by lower rectal temperature, more severe diarrhea, higher levels of OVA-specific antibodies and histamine, and altered gut microbiota composition [18]. The microbiota affects B-cell phenotype, such as antibody production (IgA, IgE, IgG1, and IgG4) and regulatory B-cell (Breg) function [20]. Dysbiosis in the gut microbiota can lead to an augmentation in the concentration of total and antigen-specific IgE. In addition to the way of delivery, breastfeeding also contributes to the development of the infant’s gut microbiome early in life [21]. Breast milk offers an abundance of sIgA along with nutrients and antimicrobial proteins from the mother. sIgA protects the infant from potential pathogens that could attack and perturb the growing gut microbiota. The maternal microbiota influences maternal sIgA repertoire, and vice versa [21]. Therefore, the acquisition and maintenance of healthy gut microbiota is a crucial aspect of FA prevention and treatment. Findings from murine models agree with the hypothesis that the development of the gut microbiota begins from the birth tract, and bacterial inoculum can only occur during breastfeeding and weaning periods [20].

R. Aitoro et al. [13] investigated the present research and prospective directions on the role of the gut microbiota in FAs and discussed the potential mechanisms by which the gut microbiota affects FA and resolution [13]. Intestinal bacteria and their metabolites, such as short-chain fatty acids (SFCA), modulate the proliferation and differentiation of T cells [14]. Promoting the differentiation from naïve T cell to regulatory T cell (Treg), and from T_H_2 skewed to T_H_1 skewed immune response, possibly through epigenetic changes on the DC or by the activation of specific G-protein coupled receptors [22]. Treg also inhibits the differentiation of naïve T cells to T_H_2 cells. The T_H_2 cell releases IL-4 and IL-13 which promote B cell CSR, from naïve B cell to IgE-producing B cell, and releases IgE triggering allergic reactions [23]. The specific mechanism of how the gut microbiota affects the occurrence of FA remains unclear and requires further investigation. Despite these findings, there remains considerable uncertainty surrounding the precise impact of gut microbiota on the occurrence of FA. Further investigation is imperative to unravel this intricate mechanism. Researchers are actively engaged in comprehending how diverse microbial compositions within our gastrointestinal tract influence immune responses associated with FA.

Current data have shown that the gut microbiota and its metabolites together with exposure to dietary and other influencing factors early in life shape infants’ immune system and tolerance to specific food antigens later in life [13]. Consequently, the manipulation of the gut microbiota using substances such as probiotics, prebiotics, and synbiotics, and techniques including fecal microbiota transplant (FMT) and diet intervention might be potential treatments and prevention for FA.

## 4. Current Knowledge of Probiotics Treatment of Food Allergy

Probiotics are living organisms that have beneficial health values when consumed in an adequate amount and may have therapeutic effects on allergic diseases and symptoms including FA [24]. The World Allergy Organization published the guidelines for allergic disease prevention, which suggested the use of probiotics for pregnant and breastfeeding women at high risk for allergy in their children. Considering all critical outcomes, there is a net benefit from the consumption of probiotics resulting in alleviation of symptoms of allergic reactions [25]. However, an important factor to be noted is that the therapeutic effects of probiotics on the immune system are strain-specific [13], meaning that some strains will be more favorable as a treatment for allergies than others. Most common probiotics come from two main bacteria genres: Lactobacillus and Bifidobacterium. These probiotics have various benefits for the gastrointestinal tract and the immune system, for instance, preventing infections, allergies, and autoimmune diseases [26].

The mechanism by which probiotics affect the gut microbiota and immune system is extremely complex. Probiotics and next-generation beneficial bacteria can influence eukaryotic cells through various mechanisms. For example, SCFAs can activate specific G-protein-coupled receptors (e.g., EPR41/43) on enteroendocrine L-cells, leading to the secretion of gut peptides that regulate energy metabolism and gut barrier functions [26]. SCFAs can also affect gene transcription by inhibiting histone deacetylase activity. In addition to SCFAs, some gut microbes communicate with the host cells through the production of other metabolites or cell components. These interactions result in diverse effects on the host, such as improving behavior in psychopathological conditions (e.g., alcoholism and autism), enhancing skin health and host metabolism through immune interaction, and the gut–brain–skin axis. Moreover, bacteria that colonize the normal microbiota, such as Barnesiella, have been associated with a lower risk of gut colonization by Vancomycin-resistant Enterococcus, while Lactobacillus treatment reduced the carriage of multi-drug-resistant potential pathogens [26].

### 4.1. Animal Trials

Animal trials are useful for exploring the mechanisms and effects of probiotics on FA, as they allow for more control over the experimental conditions and variables (Table 1). Several murine models of FA have been developed, using different food allergens, such as ovalbumin, peanut, milk, or shrimp. These models can mimic some aspects of human FA, such as IgE-mediated sensitization, anaphylactic symptoms, and histamine release. However, they also have some limitations, such as species differences, artificial induction methods, and lack of long-term follow-up.

Numerous studies have concluded positive results with the use of specific strains of probiotics as a prevention or possible treatment for FA on murine models [18,27,28,29,30,31,32]. A study found that preventive administration of a probiotic combination containing Lactobacillus and Bifidobacterium could protect CS mice from FA to OVA, by restoring the gut microbiota balance, enhancing the intestinal barrier function, inducing Treg, and modulating systemic immune responses. Studies proved that specific strains of probiotics can alleviate symptoms of FA by restoring the imbalanced T_H_1/T_H_2 immunity through the manipulation of the secretion of cytokines and chemokines towards the T_H_1-skewered immunity. X. Tian et al. [33] investigated the effects of 41 strains of probiotics on OVA-sensitized murine models. Only six strains out of the 41 strains of probiotics have shown positive effects on the alleviation of allergic reactions, which is measured and quantified by the increase in the IFN-γ/IL-4 ratio secreted by human peripheral blood mononuclear cells. Among the six strains, *Bifidobacterium animalis* KV9 (KV9) and *Lactobacillus vaginalis* FN3 (FN3) had the best results. KV9 and FN3 attenuated the allergic manifestations, mast cell degranulation, and OVA-specific IgE production in the allergic mice. The probiotics also enhanced the IFN-γ/IL-4 ratio in splenocytes and regulated the expression of TLR4, Myd88, and IRF-1 and -4 in the spleen of the allergic mice. The probiotics increased the gene expression of TLR4 and Myd88 and ameliorated FA by stimulating the TLR4 signaling pathway which modulated the T_H_1/T_H_2 cell immunology [33].

*Bifidobacterium infantis* (BB) and its antioxidant enzyme superoxide dismutase (SOD) have been experimented with to find out whether they have therapeutic effects on FA OVA-sensitized murine models. It was found that oral administration of BB significantly reduced the levels of OVA-specific IgE and IgG1, as well as the release of IL-4, IL-5, and IL-13 in the serum and splenocytes of allergic mice. BB treatment also alleviated the symptoms of anaphylaxis. BB reduced the oxidative stress in dendritic cells (DCs), as well as decreased the levels of reactive oxygen species (ROS) and malondialdehyde (MDA) and increased the levels of glutathione (GSH) and SOD in DCs. The mechanism is that BB inhibited the expression of TIM4, which is a receptor that mediates the uptake of apoptotic cells by DCs and promotes T_H_2 responses. BB also suppressed the activation of STAT6, a transcription factor that regulates TIM4 expression, by reducing its phosphorylation and nuclear translocation [34]. Other strains, including *Lactobacillus rhamnosus*, *Lactobacillus plantarum*, *Lactobacillus plantarum*, and *Pediococcus acidilactici*, etc., also showed some effects on the amelioration of symptoms induced by FA on murine models.

Despite these promising findings, there are still many challenges and limitations in animal trials on probiotics for FA. One of the major challenges is to translate the animal results to human applications as animal models may not fully reflect the complexity and heterogeneity of human FA. Therefore, animal trials should be carefully designed and validated to ensure their relevance and reproducibility for human studies. In addition, current studies have only scratched the surface and are yet to unravel the precise mechanism by which probiotics impact and treat FA. Researchers can delve deeper into comprehending this mechanism at a more microscopic level by incorporating sophisticated techniques and tools, enabling them to establish connections and pursue the intricate interaction between probiotics and the host.

In conclusion, animal trials are valuable for investigating the role of probiotics in modulating the gut microbiota and the immune system in FA. However, more research is needed to overcome their challenges and limitations and to establish their clinical relevance and applicability for human studies.

**Table 1 nutrients-15-04159-t001:** List of the probiotic strains studied in animal trials from 2015 to 2023.

Author	Year of Publication	Title	Probiotics Strain	Journal
J. Yang et al. [27]	2015	Induction of Regulatory Dendritic Cells by *Lactobacillus paracasei* L9 Prevents Allergic Sensitization to Bovine β-Lactoglobulin in Mice	*Lactobacillus paracasei* L9	J Microbiol Biotechnol
B. Yang et al. [34]	2018	Probiotics SOD inhibited food allergy via downregulation of STAT6-TIM4 signaling on DCs	*Bifidobacterium infantis*	Mol Immunol
L. Fu et al. [28]	2020	*Lactobacillus casei* Zhang Alleviates Shrimp Tropomyosin-Induced Food Allergy by Switching Antibody Isotypes through the NF-κB-Dependent Immune Tolerance	*Lactobacillus casei* Zhang	Mol Nutr Food Res
B. Y. Jin et al. [18]	2021	Probiotic Interventions Alleviate Food Allergy Symptoms Correlated With Cesarean Section: A Murine Model	Lactobacillus and Bifidobacterium	Front Immunol
C. Duan et al. [29]	2023	Oral administration of *Lactobacillus plantarum* JC7 alleviates OVA-induced murine food allergy through immunoregulation and restoring disordered intestinal microbiota	*Lactobacillus plantarum* JC7	Eur J Nutr
X. Tian et al. [33]	2023	Probiotics Alleviate Food Protein Allergy in Mice by Activating TLR4 Signaling Pathway	*Bifidobacterium animalis* KV9, *Lactobacillus vaginalis* FN3	Mol Nutr Food Res

### 4.2. Clinical Trials

Clinical trials can provide more reliable and generalizable evidence than animal trials or observational studies, as they can control for confounding factors, randomize the intervention groups, and measure the clinical outcomes (Table 2). One of the main objectives of clinical trials is to determine the optimal probiotic administration protocols for different types of FA and different populations.

B. Cukrowska et al. [35] conducted a study aimed to assess the effectiveness of a probiotic preparation containing three strains of Lactobacillus in children under 2 years of age with atopic dermatitis (AD) and cow’s milk protein allergy (CMA). The primary outcomes were changes in AD symptom severity and the proportion of children with symptom improvement. The results showed that both the probiotic and placebo groups had a significant decrease in symptom severity after the three-month intervention, which was maintained nine months later. However, the percentage of children who showed improvement was significantly higher in the probiotic group than in the placebo group after three months. Probiotics induced symptom alleviation mainly in allergen-sensitized patients, but this positive effect was not observed after nine months. The study concluded that the mixture of probiotic strains offers benefits for children with AD and CMA. However, the effect may not be sustained after the discontinuation of probiotics [35].

R. Berni Canani et al. [36] directed a study to evaluate the efficacy of an extensively hydrolyzed casein formula (EHCF) containing the probiotic *Lactobacillus rhamnosus* GG (LGG) in reducing the occurrence of other allergic manifestations (AMs) and accelerating the development of oral tolerance in children with CMA. The results showed that EHCF + LGG was effective in reducing the incidence of AMs and hastening the development of oral tolerance in children with CMA. The frequency of AMs in the probiotic group was significantly lower than that in the placebo group. The frequency of cow’s milk tolerance was significantly higher in the probiotic group than in the placebo group. The study suggested that probiotics might modulate the gut microbiota and the immune system in CMA [37]. Other studies have also suggested that combining probiotics with prebiotics (symbiotic) or with immunotherapy may enhance their effects on FA [36].

Despite these challenges and limitations, clinical trials have provided some promising evidence for the role of probiotics in preventing and treating FA. A recent systematic review and meta-analysis of randomized controlled trials (RCTs) concluded that probiotics reduced the risk of eczema by 19% in infants and children [38]. Another article concluded that probiotics improved the outcomes of oral immunotherapy (OIT) for peanut allergy by increasing the desensitization rate, reducing adverse events, and modulating the immune response [39]. However, these reviews also highlighted the heterogeneity and low quality of the included studies and called for more rigorous and standardized trials to confirm their findings.

However, it is crucial to acknowledge the limitations underscored by these reviews. The heterogeneity and substandard quality of the encompassed studies raise concerns regarding the reliability of the findings. Consequently, further rigorous and standardized trials are imperative to delve deeper into exploring the potential benefits of probiotics in managing food allergies. Overall, while there exists promising evidence supporting the utilization of probiotics for both prevention and treatment of food allergies, additional research is indispensable to definitively establish their effectiveness. By conducting meticulously designed studies with consistent methodologies, a more comprehensive understanding of how probiotics can be employed as a therapeutic option for individuals afflicted with FA can be acquired. In conclusion, clinical trials are valuable for investigating the effectiveness and safety of probiotics for FA in human populations, and more research is needed to overcome their challenges and limitations and to establish their clinical relevance and applicability for different types of FA and different populations.

**Table 2 nutrients-15-04159-t002:** List of the probiotic strains studied in clinical trials from 2001 to 2022.

Author	Year of Publication	Title	Probiotics Strain	Journal
M. Kalliomäki et al. [30]	2001	Probiotics in primary prevention of atopic disease: a randomized placebo-controlled trial	Lactobacillus GG	Lancet
A. Forsberg et al. [32]	2013	Pre- and post-natal *Lactobacillus reuteri* supplementation decreases allergen responsiveness in infancy	*Lactobacillus reuteri*	Clin Exp Allergy
D. J. Costa Et al. [40]	2014	Efficacy and safety of the probiotic *Lactobacillus paracasei* LP-33 in allergic rhinitis: a double-blind, randomized, placebo-controlled trial (GA2LEN Study)	*Lactobacillus paracasei* LP-33	Eur J Clin Nutr
R. Berni Canani et al. [41]	2016	*Lactobacillus rhamnosus* GG-supplemented formula expands butyrate-producing bacterial strains in food allergic infants	*Lactobacillus rhamnosus* GG	Isme J
R. Berni Canani et al. [37]	2017	Extensively hydrolyzed casein formula containing *Lactobacillus rhamnosus* GG reduces the occurrence of other allergic manifestations in children with cow’s milk allergy: 3-year randomized controlled trial	*Lactobacillus rhamnosus* GG	J Allergy Clin Immunol
B. Cukrowska et al. [35]	2021	The Effectiveness of Probiotic *Lactobacillus rhamnosus* and *Lactobacillus casei* Strains in Children with Atopic Dermatitis and Cow’s Milk Protein Allergy: A Multicenter, Randomized, Double Blind, Placebo Controlled Study	*Lactobacillus rhamnosus,* *Lactobacillus casei*	Nutrients
P. Loke et al. [42]	2022	Probiotic peanut oral immunotherapy versus oral immunotherapy and placebo in children with peanut allergy in Australia (PPOIT-003): a multicentre, randomised, phase 2b trial	*Lactobacillus rhamnosus* ATCC 53103	Lancet Child Adolesc Health

## 5. Future of Probiotics as a Treatment for Food Allergy

### 5.1. Multi-Strain and Synbiotic Formulations for Food Allergy

Probiotics have been shown to influence various aspects of the immune system, such as cytokine production, antibody secretion, antigen presentation, and regulatory cell function [43]. These effects may help to reduce the allergic inflammation and promote the immune tolerance to food antigens. Some of the most studied probiotic strains for FA belong to the genera Lactobacillus and Bifidobacterium. These strains have been reported to alleviate the allergic symptoms and histamine release in animal models of FA, as well as to modulate the antibody isotype switching, cytokine production, and antigen presentation in humans and animals [44].

In addition to single-strain probiotics, multi-strain or synbiotic (probiotics combined with prebiotics) formulations may also have beneficial effects for FA [45]. For example, a recent study showed that a synbiotic mixture containing four strains of Bifidobacterium and Lactobacillus and a prebiotic oligosaccharide reduced the incidence and severity of cow’s milk allergy in infants at high risk. Another study showed that a multi-strain probiotic containing seven strains of Lactobacillus, Bifidobacterium, and Streptococcus enhanced the oral immunotherapy outcomes in children with peanut allergy. These studies suggest that combining different probiotic strains or adding prebiotics may enhance the synergistic or additive effects of probiotics for FA [46].

### 5.2. Precision Therapy with Synthetic Biology for Food Allergy

A considerable amount of research has placed significant emphasis on the exploration of novel strains of probiotics and their potential impact on alleviating symptoms associated with FA. However, it is anticipated that the market will eventually become saturated and chaotic due to an overwhelming number of available choices. To propel further development, researchers must devise innovative approaches to narrow down these possibilities. This review aims to propose two potential avenues: (1) transitioning from ambiguous treatment methods towards precise and targeted interventions; (2) shifting the focus from discovering new strains to modifying existing ones.

Precision therapy is a medical treatment that is adapted to each patient’s specific characteristics, such as their genes, proteins, and other biomolecules. Precision therapy aims to provide the right treatments to the right patients at the right time, and to improve the effectiveness and safety of the therapies [7]. Precision therapy approaches have attracted significant investment in the past decade to develop new therapies, understand disease mechanisms, and possibly prevent diseases before they occur. However, precision therapy investments may compromise existing public health interventions that could have a larger effect on population health in many aspects [47]. The assessment of the human epidermal growth factor receptor (HER)-2 status in breast cancer patients is a well-known example of this approach. HER-2 was initially identified as a prognostic factor that indicated a higher likelihood of a more aggressive disease progression in positive patients [48]. As for FA, perhaps researchers can find the growth factor or other biomarkers responsible for the cause of allergic reactions in most of the allergic population with the help of synthetic biology to alter the genetic information of the probiotics, using techniques such as plasmid transfection to target and block the expression of the FA biomarkers in patients (Figure 3).

Synthetic biology has received increasing attention over the past few decades. Synthetic biology is a field of science that applies engineering principles to design and modify living systems and organisms [6]. The concept has been incorporated to drive biological applications and discovery [49] and to solve problems in medicine, manufacturing, and agriculture. What can synthetic biology do? How can researchers integrate the idea into studying probiotics as a treatment for FA? These can be some questions to consider when executing future research on probiotics and FA. One of the potential applications of synthetic biology can be to artificially modify the genetic material of probiotic strains with the purpose of stabilizing or escalating the production of the metabolite(s) responsible for the alleviation of allergic symptoms. Hypothetically, if the metabolite(s) can be identified to have therapeutic effects, researchers can then (1) genetically modify the bacterial strain to allow stabilized or elevated production of the compound(s) and (2) isolate the compound(s) to allow in vitro mass production. In a study of colorectal tumorigenesis, Q. Zhang et al. have identified indole-3-lactic acid, which is a metabolite of the probiotics strain Lactobacillus plantarum and discovered that it has the same therapeutic effects on diseased mice as just the strain itself [50].

The development of genetic modification and the improvement of the ability to synthesize artificial genomes have promoted the expansion of the depth of artificial genome design. Techniques of artificial genome synthetics include the simplification and deletion of non-essential genes and genetic elements, and the modification of codons [51]. Current studies and research have achieved the conversion of eighteen thousand target codons in *Escherichia coli*, and have constructed an artificial strain with only 61 genetic codons [52]. L. Sanmarco et al. discovered that lactate can limit T cell-driven autoimmune diseases by activating the expression of NDUFA4LA, which controls the mitochondrial activity of DCs and the ROS-driven transcription program (the program promotes T cell differentiation) through a hypoxia-inducible factor-1α (HIF-1α)-driven mechanism. An engineered probiotic Ecn^Lac^ was designed for the stabilized production of lactate from pyruvate, targeting the activation of HIF-1α-NDUFA4L2 signaling. The introduction of the *IdhA* gene by plasmid transfection and the knockout of the *pta* gene in *Escherichia coli* Nissle (EcN) ensured the conversion from pyruvate to D-lactate [53]. The concept of synthetic biology and studies from other research fields can be inspirational and offer researchers an innovative perspective on the advancement of the field of FA. Synthetic biology and precision therapy may offer new tools and strategies to engineer probiotics with improved efficacy and safety for FA treatment.

## 6. Discussion and Conclusions

Despite these promising findings, there are still many challenges and limitations in the field of probiotics for FA. One of the major challenges is to identify the optimal strain(s), dose(s), duration(s), and timing(s) of probiotic administration for different types of FA and different populations. Another challenge is to elucidate the molecular mechanisms by which probiotics interact with the host cells and the gut microbiota, and how these interactions affect the immune response to food antigens and possibly investigate the mechanisms on an epigenetics perspective. A third challenge is to ensure the quality, safety, and efficacy of probiotic products, as well as their standardization and regulation. A fourth challenge is to conduct well-designed clinical trials with adequate sample size, duration, follow-up, outcome measures, and reporting standards. Additionally, the quantification of symptoms in FA should incorporate more advanced and precise biomarkers. For instance, recent discoveries have shown a correlation between exhaled nitric oxide (FeNO) and FA [54]. FeNO is a gaseous byproduct generated by the inflamed or irritated cells lining the airways in response to allergens. FeNO can be quantified through a straightforward breath test, serving as an indicator of airway inflammation. Elevated levels of FeNO suggest heightened inflammation, potentially indicating allergic asthma or other allergic conditions. The utilization of FeNO testing aids in diagnosing and monitoring allergic disorders while guiding treatment strategies involving anti-inflammatory medications [55].

On the contrary, the advancement and promotion of genetically modified probiotics encounter numerous obstacles and challenges. Primarily, engineered probiotics may give rise to potential risks and ethical dilemmas, such as gene transfer, horizontal gene transfer, biosafety concerns, and public acceptance. Additionally, they might induce unforeseen adverse effects on both the host organism and the surrounding environment, thus necessitating further investigation through clinical trials to establish their long-term consequences. Moreover, regulation and legality pertaining to this field remain ambiguous in most countries. In the United States specifically, there are presently no regulations governing probiotics or engineered probiotics as a distinct regulatory product category under either the Federal Food, Drug, and Cosmetic Act (FD&C Act) or the Public Health Service Act (PHSA). In China, a similar situation prevails with regards to probiotics and engineered probiotics lacking clear regulatory guidelines as a distinct product category. Moreover, the classification of engineered probiotics necessitates careful deliberation due to their multidisciplinary nature. Determining the boundaries for categorizing engineered probiotics is challenging since it involves the convergence of various fields. Presently, genetically modified probiotics remain an emerging area of research with limited animal trials and no clinical trials conducted thus far. The efficacy and potential side effects of engineered probiotics remain uncertain due to the current state of knowledge. The promotion of clinical trials and marketing of these products in later stages still face numerous challenges and uncertainties that need to be overcome. The absence of definitive and uniform guidelines and policies may impede the advancement and commercialization of engineered probiotics. Despite the difficulties ahead, ongoing research is imperative, while laws and regulations should keep pace with the advancements in demand for these products to ensure their safe development.

In conclusion, probiotics may have a potential role in preventing and treating FA by modulating the gut microbiota and the immune system. However, more research is needed to establish their clinical effectiveness and safety, as well as their underlying mechanisms of action. Probiotics may not be a magic bullet for FA, but they may be a valuable adjunct or alternative to conventional therapies in some cases.

## Figures and Tables

**Figure 1 nutrients-15-04159-f001:**
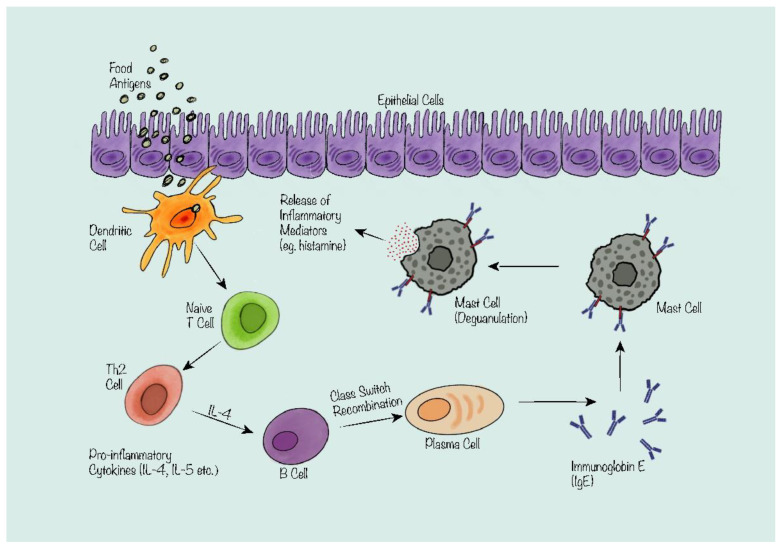
Overview of the mechanism of IgE-mediated food allergy. Antigen-presenting cells (e.g., dendritic cells) process and present the antigens to lymphocytes (e.g., T cell). Naïve T cell differentiates into T_H_2 cell and releases pro-inflammatory cytokines such as IL-4 and IL-5. IL-4 stimulates B cell class switch recombination to produce IgE. IgE binds to the high-affinity IgE receptors on immune effector cells (e.g., mast cells). Mast cell degranulates and releases inflammatory mediators such as histamine which results in the symptoms of food allergy.

**Figure 2 nutrients-15-04159-f002:**
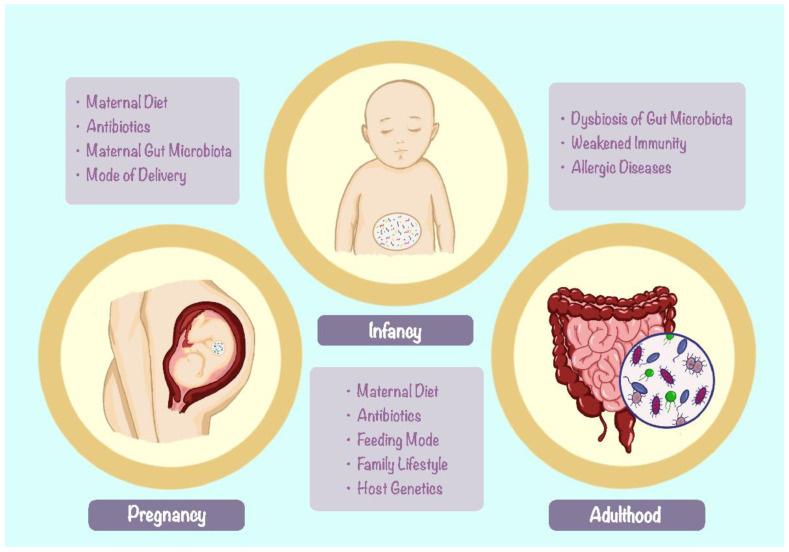
Overview of factors influencing the development of gut microbiota throughout the three periods of life: pregnancy, infancy, and adulthood.

**Figure 3 nutrients-15-04159-f003:**
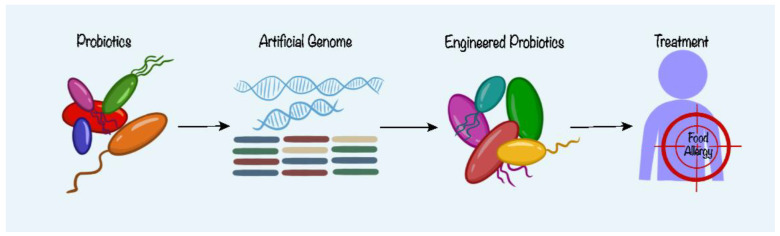
Potential approach to design engineered probiotics for the targeted treatment of food allergy. Modification of probiotics through the building of artificial genome to target specific pathways affecting the occurrence of food allergy.

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
