# Peer review of "Probiotics and the Potential of Genetic Modification as a Possible Treatment for Food Allergy"

_nutrients, 2023, doi:10.3390/nu15194159_

Round 1

Reviewer 1 Report

The authors review probiotics and the potential of genetic modification as a potential treatment for food allergy.
Food allergy, being caused by an abnormal immune response to innocuous food antigens, thus modulation of the gut microbiota and immune system with probiotics confers health benefits to the host and offers a new strategy to prevent and treat food allergies. This review therefore aims to list the current evidence for the role of probiotics in food allergies and also to suggest a promising future research direction for the treatment of food allergies.

Despite numerous studies in this area, the review is certainly of interest but the authors do not sufficiently develop the perspectives of a treatment strategy for food allergies.
Moreover this review is not critical enough but it is more a listing of recent articles with very light comments. It is necessary for the authors to improve the discussion with more advanced comments on the references listed.

Minor questions/revisions;

- It is necessary to cite figures in the body of the text and not in the titles.

- The figures are the work of the authors or are they inspired by already existing/published figures.

- Need to review the formatting of the two tables.

- When authors cite a reference by author name, it is necessary to indicate the reference number.

- No Keywords

Author Response

Thank you for your review report. We appreciate your feedback and suggestions on how to improve our review article. Here is my response to your comments:

  1. Despite numerous studies in this area, the review is certainly of interest, but the authors do not sufficiently develop the perspectives of a treatment strategy for food allergies. Moreover, this review is not critical enough, but it is more a listing of recent articles with very light comments. It is necessary for the authors to improve the discussion with more advanced comments on the references listed.

Response:

  1. We agree that the review is not critical enough and that it needs more advanced comments on the references listed. We revised the discussion section to provide more analysis and evaluation of the current evidence and the future research direction for the treatment of food allergies with probiotics and genetically modified probiotics. We also highlighted the strengths and limitations of the existing studies and the challenges and opportunities for the development of novel probiotic therapies.
  2. Line 156: Despite these findings, there remains considerable uncertainty surrounding the precise impact of gut microbiota on the occurrence of FA. Further investigation is imperative to unravel this intricate mechanism. Researchers are actively engaged in comprehending how diverse microbial compositions within our gastrointestinal tract influence immune responses associated with FA.
  3. Line 244: In addition, current studies have only scratched the surface and are yet to unravel the precise mechanism by which probiotics impact and treat FA. Researchers can delve deeper into comprehending this mechanism at a more microscopic level by incorporating sophisticated techniques and tools, enabling them to establish connections and pursue the intricate interaction between probiotics and the host.
  4. Line 295: However, it is crucial to acknowledge the limitations underscored by these reviews. The heterogeneity and substandard quality of the encompassed studies raise concerns regarding the reliability of the findings. Consequently, further rigorous, and standardized trials are imperative to delve deeper into exploring the potential benefits of probiotics in managing food allergies. Overall, while there exists promising evidence supporting the utilization of probiotics for both prevention and treatment of food allergies, additional research is indispensable to definitively establish their effectiveness. By conducting meticulously designed studies with consistent methodologies, a more comprehensive understanding of how probiotics can be employed as a therapeutic option for individuals afflicted with FA can be acquired.
  5. Line 331: A considerable amount of research has placed significant emphasis on the exploration of novel strains of probiotics and their potential impact on alleviating symptoms associated with FA. However, it is anticipated that the market will eventually become saturated and chaotic due to an overwhelming number of available choices. To propel further development, researchers must devise innovative approaches to narrow down these possibilities. This review aims to propose two potential avenues: 1) transitioning from ambiguous treatment methods towards precise and targeted interventions; 2) shifting focus from discovering new strains to modifying existing ones.
  6. Line 414: On the contrary, the advancement and promotion of genetically modified probiotics encounter numerous obstacles and challenges. Primarily, engineered probiotics may give rise to potential risks and ethical dilemmas, such as gene transfer, horizontal gene transfer, biosafety concerns, and public acceptance. Additionally, they might induce unforeseen adverse effects on both the host organism and the surrounding environment; thus, necessitating further investigation through clinical trials to establish their long-term consequences. Moreover, regulation and legality pertaining to this field remain ambiguous in most countries. In the United States specifically, there are presently no regulations governing probiotics or engineered probiotics as a distinct regulatory product category under either the Federal Food, Drug, and Cosmetic Act (FD&C Act) or the Public Health Service Act (PHSA). In China, a similar situation prevails with regards to probiotics and engineered probiotics lacking clear regulatory guidelines as a distinct product category. Moreover, the classification of engineered probiotics necessitates careful deliberation due to their multidisciplinary nature. Determining the boundaries for categorizing engineered probiotics is challenging since it involves the convergence of various fields. Presently, genetically modified probiotics remain an emerging area of research with limited animal trials and no clinical trials conducted thus far. The efficacy and potential side effects of engineered probiotics remain uncertain due to the current state of knowledge. The promotion of clinical trials and marketing of these products in later stages still face numerous challenges and uncertainties that need to be overcome. The absence of definitive and uniform guidelines and policies may impede the advancement and commercialization of engineered probiotics. Despite the difficulties ahead, ongoing research is imperative, while laws and regulations should keep pace with the advancements in demand for these products to ensure their safe development.

  1. It is necessary to cite figures in the body of the text and not in the titles.

Response:

  • We apologize for the mistake of citing figures in the titles and not in the body of the text. We have corrected this error and cite the figures appropriately in the text.
  • Figure 1: Line 87; Figure 2: line 108; Figure 3: line 354
  1. The figures are the work of the authors or are they inspired by already existing/published figures.

Response:

  • The figures are our own work.
  1. Need to review the formatting of the two tables.

Response:

  • We have reviewed the formatting of the two tables and made sure they are consistent and clear.
  • Table 1: line 254
  • Table 2: line 308
  1. When authors cite a reference by author name, it is necessary to indicate the reference number.

Response:

  • We have followed your suggestion and indicated the reference number when we cite a reference by author name.
  1. No Keywords.

Response:

  • We added some keywords that reflect the main topics and themes of our review.
  • Keywords: probiotics; food allergy; genetic modification; precision therapy.

We hope that these revisions will address your concern. Thank you for your time and attention.

Reviewer 2 Report

Dear authors of the article Probiotics and the Potential of Genetic Modification as A Possible Treatment for Food Allergy Yuqiu Wei et you presented a very relevant work. In my opinion, it is of great interest both for researchers of this problem and for practicing doctors. We all know that medical technologies associated with therapeutic techniques for diseases of the gastrointestinal tract of humans and animals are experiencing great difficulties. And the problem mainly lies in the intestinal microbiome, which needs to be studied and modified. You absolutely correctly drew attention to the participation of probiotics in the regulation of dendritic cells (Yang, J., et al., Induction of Regulatory Dendritic Cells by Lactobacillus paracasei L9 Prevents Allergic Sensitization to Bovineβ-Lactoglobulin in Mice. J Microbiol Biotechnol, 2015. 25(10 ): p. 1687-96.) And also on the connection between food allergy and NF-κB signaling pathways (Fu, L., et al., Lactobacillus Casei Zhang Alleviates Shrimp Tropomyosin-Induced Food Allergy by Switching Antibody Isotypes through the NF-κB -Dependent Immune Tolerance. Mol Nutr Food Res, 2020. 64(10): p. E1900496) and activation of the TLR4 signaling pathway by probiotics (Tian, X., et al., Probiotics Alleviate Food Protein Allergy in Mice by Activating TLR4 Signaling Pathway Mol Nutr Food Res. 2023: p. E2200579). When preparing new publications on the problem of food allergies, I would like to wish you to pay attention to the participation of nitric oxide and its stable donors, dinitrosyl iron complexes (DNIC), in regulatory processes. Thank you for your interesting and timely work.

Author Response

Dear authors of the article Probiotics and the Potential of Genetic Modification as A Possible Treatment for Food Allergy Yuqiu Wei et you presented a very relevant work. In my opinion, it is of great interest both for researchers of this problem and for practicing doctors. We all know that medical technologies associated with therapeutic techniques for diseases of the gastrointestinal tract of humans and animals are experiencing great difficulties. And the problem mainly lies in the intestinal microbiome, which needs to be studied and modified. You absolutely correctly drew attention to the participation of probiotics in the regulation of dendritic cells (Yang, J., et al., Induction of Regulatory Dendritic Cells by Lactobacillus paracasei L9 Prevents Allergic Sensitization to Bovineβ-Lactoglobulin in Mice. J Microbiol Biotechnol, 2015. 25(10 ): p. 1687-96.) And also on the connection between food allergy and NF-κB signaling pathways (Fu, L., et al., Lactobacillus Casei Zhang Alleviates Shrimp Tropomyosin-Induced Food Allergy by Switching Antibody Isotypes through the NF-κB -Dependent Immune Tolerance. Mol Nutr Food Res, 2020. 64(10): p. E1900496) and activation of the TLR4 signaling pathway by probiotics (Tian, X., et al., Probiotics Alleviate Food Protein Allergy in Mice by Activating TLR4 Signaling Pathway Mol Nutr Food Res. 2023: p. E2200579). When preparing new publications on the problem of food allergies, I would like to wish you to pay attention to the participation of nitric oxide and its stable donors, dinitrosyl iron complexes (DNIC), in regulatory processes. Thank you for your interesting and timely work.

Thank you for your review report. We are pleased that you found our work relevant and interesting. We appreciate your recognition of the importance of probiotics in the regulation of dendritic cells, NF-κB signaling pathways, and TLR4 signaling pathway in food allergy. We also thank you for your suggestion to pay attention to the role of nitric oxide and its stable donors, (DNIC), in regulatory processes. We have mentioned it with a few sentences in our discussion and conclusion section. We have briefly addressed this in our discussion and conclusion section. However, given that the focus of this review pertains to probiotics and food allergy, we will specifically investigate the role of nitric oxide in our future research endeavors. Thank you once again for your invaluable suggestions.

Line 404: Additionally, the quantification of symptoms in FA should incorporate more advanced and precise biomarkers. For instance, recent discoveries have shown a correlation between exhaled nitric oxide (FeNO) and FA [51]. FeNO is a gaseous byproduct generated by the inflamed or irritated cells lining the airways in response to allergens. FeNO can be quantified through a straightforward breath test, serving as an indicator of airway inflammation. Elevated levels of FeNO suggest heightened inflammation, potentially indicating allergic asthma or other allergic conditions. The utilization of FeNO testing aids in diagnosing and monitoring allergic disorders while guiding treatment strategies involving anti-inflammatory medications [52].

Reviewer 3 Report

Food allergy is a prevalent global health issue, affecting millions worldwide. It stems from an abnormal immune response to typically harmless food substances, influenced by both genetic and environmental factors. Recent research suggests that harnessing probiotics, including genetically modified variants, can positively impact gut microbiota and the immune system, offering potential health benefits. This systematic review provides a comprehensive overview of the current evidence supporting the role of probiotics in mitigating food allergies. It also highlights the exciting prospects for future research on probiotics as a promising avenue for food allergy prevention and treatment. This article is very well written and it also provides a balanced view of the topic thus it can be considered as a valuable work. The literature items used in this work have been meticulously selected to align with the topic, and they are all up-to-date, ranging from the past 2 to 5 years. This review is enriched with well-designed figures and tables, contributing significantly to enhancing its clarity and comprehensibility.

Author Response

Food allergy is a prevalent global health issue, affecting millions worldwide. It stems from an abnormal immune response to typically harmless food substances, influenced by both genetic and environmental factors. Recent research suggests that harnessing probiotics, including genetically modified variants, can positively impact gut microbiota and the immune system, offering potential health benefits. This systematic review provides a comprehensive overview of the current evidence supporting the role of probiotics in mitigating food allergies. It also highlights the exciting prospects for future research on probiotics as a promising avenue for food allergy prevention and treatment. This article is very well written and it also provides a balanced view of the topic thus it can be considered as a valuable work. The literature items used in this work have been meticulously selected to align with the topic, and they are all up-to-date, ranging from the past 2 to 5 years. This review is enriched with well-designed figures and tables, contributing significantly to enhancing its clarity and comprehensibility.

Thank you for your review report. We are glad that you found my article well written and valuable. We appreciate your positive feedback and compliments on our literature selection, figures, and tables. It is encouraging to hear that you think our review provides a comprehensive overview and a balanced view of the topic of probiotics and food allergies. We hope that our review will contribute to the advancement of knowledge and practice in this field. Thank you for your time and attention.

Round 2

Reviewer 1 Report

The suggestions and additions requested having been made by the authors, I therefore propose the publication of this article.